Slovak morphological tokenizer using the Byte-Pair Encoding algorithm

http://orcid.org/0000-0002-1878-577X Držík Dávid david.drzik@ukf.sk
Forgac Frantisek
Department of Informatics, Constantine the Philosopher University in Nitra , Nitra , Slovak Republic
Benítez-Andrades José Alberto
Electronic publication date: 2024 Nov 19
Publication date: 2024
Volume: 10
Electronic Location ID: e2465
Received 2024 May 2; Accepted 2024 Oct 8
Copyright: © 2024 Držík and Forgac
Copyright year: 2024
Copyright holder: Držík and Forgac
License: This is an open access article distributed under the terms of the Creative Commons Attribution License, which permits unrestricted use, distribution, reproduction and adaptation in any medium and for any purpose provided that it is properly attributed. For attribution, the original author(s), title, publication source (PeerJ Computer Science) and either DOI or URL of the article must be cited.
License URL: https://creativecommons.org/licenses/by/4.0/

Keywords: Morphological tokenization, Slovak language, Word root integrity, Language model

Funding: Scientific Grant Agency of the Ministry of Education of the Slovak Republic and Slovak Academy of Sciences VEGA-1/0821/21 Slovak Research and Development Agency APVV-18-0473 National Project National Competence Centre for High Performance Computing 11070AKF2 European Regional Development Fund EU Structural Funds Informatization of society, Operational Program Integrated Infrastructure This work was supported by the Scientific Grant Agency of the Ministry of Education of the Slovak Republic and Slovak Academy of Sciences under Contract VEGA-1/0821/21, also by the Slovak Research and Development Agency under the contract no. APVV-18-0473. Part of the research results were obtained using the computational resources procured in the national project National Competence Centre for High Performance Computing (project code: 311070AKF2) funded by European Regional Development Fund, EU Structural Funds Informatization of society, Operational Program Integrated Infrastructure. The funders had no role in study design, data collection and analysis, decision to publish, or preparation of the manuscript.

==============================
This study introduces a new approach to text tokenization, SlovaK Morphological Tokenizer (SKMT), which integrates the morphology of the Slovak language into the training process using the Byte-Pair Encoding (BPE) algorithm. Unlike conventional tokenizers, SKMT focuses on preserving the integrity of word roots in individual tokens, crucial for maintaining lexical meaning. The methodology involves segmenting and extracting word roots from morphological dictionaries and databases, followed by corpus preprocessing and training SKMT alongside a traditional BPE tokenizer. Comparative evaluation against existing tokenizers demonstrates SKMT’s outstanding ability to maintain root integrity, achieving 99.7% root integrity compared to SlovakBERT (90.5%) and a pureBPE tokenizer (93.1%). Further validation involved fine-tuning models on a sentiment classification NLP task, where models trained with SKMT achieved an F1-score improvement of 3.5% over those trained with conventional BPE tokenization, followed by a focus on the Semantic Textual Similarity (STS) task. These findings suggest that training language models on the SKMT tokenizer significantly enhances model performance and quality.

Introduction

The aim of natural language processing (NLP) is to enable machine learning algorithms to effectively process textual data and understand language as humans do. Currently, this field is often associated with the development and training of large language models, which have the ability to address various NLP tasks such as machine translation, sentiment analysis, text generation, and many others. Before training a language model, text needs to be transformed into a format that the model can process. Most language models nowadays use subword tokenization to achieve this, which divides text into tokens represented by numerical identifiers. The main motivation for subword tokenization is to provide the model with a sequence of meaningful units while limiting the size of the vocabulary, ensuring that the representation captures as much of the meaning as possible without becoming unwieldy. Among the most well-known algorithms for subword tokenization are Byte-Pair Encoding (BPE) (Sennrich, Haddow & Birch, 2016) and WordPiece (Wu et al., 2016). Such algorithms are based on statistical principles and do not take into account the morphological level of language, which may result in limitations in understanding language structures and their meanings.

In this article, we introduce a new approach to text tokenization for the Slovak language based on the BPE algorithm, which we call the SlovaK Morphological Tokenizer (SKMT). The goal is to utilize the basic lexical unit (word root) that carries the meaning of the word when training the BPE tokenizer algorithm. We believe that incorporating such root morphemes can improve tokenization, potentially leading to more precise training of language models.

We compare our trained tokenizer with the existing tokenizer of the SlovakBERT language model (Pikuliak et al., 2022), focusing mainly on the ability to preserve the integrity of word roots in the resulting tokens. In the following example (Fig. 1), we demonstrate a tokenized sentence using the SlovakBERT tokenizer. Sentence: “Farboslepú červenovlásku ohrozili protiidúce autá, a preto núdzovo zaparkovala svoje auto v močarine.” and its translation: “The color-blind redhead was threatened by oncoming cars, so she parked her car in a swamp.” Word roots that have been divided into multiple tokens are highlighted in red and underlined, while word roots that remain intact within a single token are highlighted in green and bold without underlining.

Figure 1 Demonstration of tokenization by tokenizer SlovakBERT.

The rest of this article is structured as follows: In “Related works”, we discuss current tokenization methods and similar works that have attempted to incorporate morphology into text tokenization. “Materials and Methods” describes the acquisition of root morphemes of Slovak words and the subsequent training of our morphological tokenizer using the BPE algorithm. In “Results”, we compare the tokenization results with the tokenizers of the SlovakBERT model and with a trained pure BPE algorithm. Additionally, we present the fine-tuning of models and evaluation on the sentiment classification task and the Semantic Textual Similarity task, demonstrating the enhanced performance achieved with the SKMT tokenizer.

Related work

Byte-Pair Encoding (BPE) is a widely used method for developing and implementing an encoding strategy for natural language texts. It is among the most prevalent tokenization techniques employed in language models (Sennrich, Haddow & Birch, 2016; Radford et al., 2019; Bostrom & Durrett, 2020; Brown et al., 2020; Le Scao et al., 2023), as well as in various other language modeling applications, such as machine translation (Ding, Renduchintala & Duh, 2019) and chatbots (Zhang et al., 2020).

Mager et al. (2022) examined the impact of different tokenization methods on Transformer-based language models. They found that standard tokenization methods simplify the true linguistic structure of words, which consists of meaningful morphemes. Their study compared BPE with two morphological algorithms (Morfessor and StateMorph), focusing on model perplexity, training efficiency, and performance on NLP tasks. Using languages with rich morphology like Finnish, Turkish, and Russian, as well as English for comparison, they pre-trained GPT and BERT models with each method. Results showed that morphological tokenization achieved lower perplexity, faster convergence, and often better performance in NLP tasks compared to BPE.

Semi-Supervised Learning of Concatenative Morphology presents an extension of the Morfessor Baseline model for morphological segmentation to a semi-supervised learning context. By incorporating a small set of labeled linguistic data into the training process alongside a larger set of unlabeled data, the authors demonstrate significant improvements in morphological segmentation for both English and Finnish. The semi-supervised approach introduces weights to balance the influence of labeled and unlabeled data, optimizing the segmentation accuracy. The results show that even minimal labeled data can enhance performance, surpassing state-of-the-art unsupervised methods. This methodology not only advances the field of morphological analysis but also holds potential benefits for various downstream NLP tasks (Kohonen, Virpioja & Lagus, 2010; Smit et al., 2014).

Zouhar et al. (2023) delve into the complexities and formal foundations of Byte-Pair Encoding as a combinatorial optimization problem. This is a significant theoretical advancement as it helps clarify the operational principles and optimization objectives of BPE, which were not fully understood previously. They analyze BPE using submodular functions and demonstrate that the iterative greedy approach of BPE approximates an optimal solution. Specifically, they quantify this approximation and provide a mathematical bound on its efficiency, thereby offering a clearer understanding of the algorithm’s performance limits. The article introduces a more efficient implementation of the BPE algorithm that significantly reduces its computational complexity over the original implementation (Gage, 1994). The authors improve the runtime from O(NM) to O(NlogM), where N is the sequence length and M is the merge count. This optimization could lead to faster processing times in practical applications, particularly in large-scale NLP tasks. Furthermore, they explore optimizations to the original brute-force approach used for finding the optimal BPE merges. By incorporating memorization techniques, they enhance the efficiency of finding the best merge sequence, which can be especially useful in settings where maximum compression or tokenization efficiency is desired.

The proof of correctness for the proposed methodologies by Bostrom & Durrett (2020) relies heavily on the theoretical framework provided by the theory of submodular functions. Submodular functions, a cornerstone in combinatorial optimization, are instrumental in modeling and optimizing various types of resource allocation problems, including those related to data compression and natural language processing tasks. As detailed by Krause & Golovin (2014) and more recently expanded upon by Bilmes (2022), submodular functions offer a robust mathematical approach to ensure that the algorithms yield near-optimal solutions under certain greedy conditions. In this context, they were able to establish that the utility derived from data compression, which is central to their study, constitutes a special kind of submodular function over a constrained optimization space, echoing foundational insights by Malekian (2009). This theoretical underpinning not only supports the efficacy of the proposed methods but also aligns with broader efforts to apply advanced mathematical principles to improve computational techniques in data processing and machine learning.

Bostrom & Durrett (2020) begin by outlining the foundational aspects of BPE and unigram LM tokenization. BPE, a method popularized for its efficacy in managing large vocabularies and reducing out-of-vocabulary words, operates by merging the most frequent pairs of characters or character sequences iteratively. Conversely, unigram LM tokenization approaches the segmentation of text probabilistically, optimizing a likelihood function to choose the best segmentation over the entire corpus. The core of their investigation involves a series of empirical tests to measure the performance of transformer language models pretrained with these tokenization methods across several NLP tasks, including text classification and machine translation, in both English and Japanese. The choice of languages, one with simpler morphological structures (English) and one more complex (Japanese), is particularly pertinent to evaluating the robustness of the tokenization methods under varying linguistic conditions.

The study’s findings are revealing unigram LM tokenization not only aligns better with morphological rules but also mitigates some of the inherent problems in BPE’s greedy merging strategy. Specifically, models using unigram LM tokenization demonstrated improved performance over those using BPE, with particular gains noted in tasks involving morphological complexity. This suggests that the unigram approach could lead to better generalization in real-world applications across different languages and tasks. Bostrom & Durrett (2020) propose a revision of the conventional methodologies employed in language model pretraining. They suggest that researchers and practitioners in natural language processing should adopt unigram language model (LM) tokenization instead of byte pair encoding (BPE), highlighting its enhanced performance and superior alignment with linguistic structures. This recommendation is poised to influence future research directions and pretraining practices in the NLP community. The study provides a critical evaluation of tokenization methods in the context of NLP model development by highlighting the limitations of BPE and demonstrating the advantages of unigram LM tokenization, this article contributes significantly to the ongoing discussion about optimizing the preprocessing steps critical to successful model training.

Tacorda et al. (2017) extended the BPE algorithm to what they called Controlled Byte-Pair Encoding (CBPE). Their approach was specifically tailored to better handle morphological segmentation in neural machine translation systems. CBPE prevents BPE from merging morphologically significant affixes with root words indiscriminately. This controlled merging significantly improved the handling of inflected forms in morphologically rich languages, as demonstrated in their translation models for English to Filipino, a language known for its complex verbal inflections.

In the study by Gutierrez-Vasques, Bentz & Samardžić (2023), BPE is examined morphologically. The research covers BPE’s application in 47 typologically varied languages and its impact on text compression. A notable discovery is that subwords identified during BPE’s initial merging stages significantly affect compression, especially in languages with complex inflectional morphology, capturing subwords that reflect morphological characteristics.

Contrary to the common perception that BPE subwords are linguistically irrelevant, Gutierrez-Vasques, Bentz & Samardžić (2023) show that these compression patterns resemble traditional morphological structures across different languages. This finding disputes the idea of BPE as simply a technical device, instead suggesting it can reveal linguistic insights. They introduce a new way to categorize languages by their BPE subword traits, which utilizes BPE’s accidental linguistic correlations to assess morphological productivity without needing annotated resources or linguistic expertise.

Their findings suggest that understanding the relationship between BPE and linguistic morphology could greatly improve language processing in multilingual NLP settings. This has significant implications for both quantitative typology and the creation of linguistically aware NLP tools, redefining BPE as both a practical tool and a method for linguistic examination.

In practical terms, their method not only offers a quantitative mechanism for analyzing morphological typology but also holds potential benefits for various NLP tasks. For instance, several NLP applications utilize typological language vectors such as lang2vec for diverse purposes, and our approach generates similar language vectors. This presents an opportunity to enhance the current typological vectors with morphological insights in a cost-effective manner.

This is particularly promising for transfer learning in highly multilingual environments, where the efficacy of the transfer hinges on factors like language similarity and the availability of annotated resources (cited in studies by Lauscher et al. (2020) and Pires, Schlinger & Garrette (2019)). Indeed, identifying which languages are best suited for transfer is an ongoing research issue (highlighted by Lin et al. (2019) and Malkin, Limisiewicz & Stanovsky (2022)). Their approach offers a new, informed criterion to aid cross-linguistic transfer, especially in challenging situations such as zero-shot cross-lingual model transfers between under-resourced and linguistically distant languages.

Materials and Methods

The goal of this contribution is to introduce our proposed method of tokenization, which utilizes morphology in training the tokenizer using the BPE algorithm. The majority of existing tokenizers do not take into account the morphological level of the language and tokenize text solely based on frequency of occurrence. This often leads to splitting the root of a word into multiple tokens. Since the root of a word carries the lexical meaning of the word, it should not be split into multiple tokens (Sennrich, Haddow & Birch, 2016). Our motivation is to create a tokenizer that can preserve the integrity of roots in individual tokens. We assume that this approach will be significant in the development of language models, where it is important to maintain the integrity of word roots for better understanding and accuracy in text processing. Additionally, we analyze fundamental statistics, such as the count of unique valid words in the text or the number of unique words with recorded roots, across three tokenizers’ tested tokenization properties. These metrics aim to offer valuable insights into how effectively and efficiently they process text. Through the analysis of the results, we aim to determine whether our proposed method of tokenization can preserve the integrity of roots in tokens with higher accuracy compared to the existing BPE tokenizer, which was trained in the traditional way. The individual methodological steps are shown in Fig. 2 and described below.

Figure 2 Methodological steps.

The entire methodology of the practical part of the research can be summarized into the following steps: Segmentation of root morphemes from Slovak morphemic dictionaries (RMDS and DRMS) and extraction of word roots from the morphological database SNC (MDB SNC) (Slovak National Corpus, 2024).

Selection of a text corpus and its preprocessing.

Design and training of the SlovaK Morphological Tokenizer (SKMT).

Training of the pureBPE tokenizer, which uses an optimized version of the BPE algorithm from Hugging Face, on the same text corpus as SKMT.

Tokenization of a random text sample consisting of 500,000 paragraphs with three tokenizers, including an existing language model tokenizer SlovakBERT, pureBPE, and our morphological tokenizer SKMT.

Evaluation of tokenization results using six characteristics (Total Tokens (TT), Average Tokens per Word (AT/W), Average Tokens per Rooted Word (AT/RW), Average Token Length (ATL), Average Rooted Token Length (ARTL), Correct Tokenization (CT)) and comparison of our SKMT tokenizer with tokenizers created using the traditional approach (SlovakBERT and pureBPE).

Pretraining language models on texts tokenized by both the SKMT and pureBPE tokenizers, followed by fine-tuning on NLP tasks such as sentiment classification and Semantic Textual Similarity (STS), to compare the performance improvements.

Segmentation and extraction of word roots

To create our own morphological tokenizer capable of preserving the integrity of roots within individual tokens, it is necessary to first obtain roots for as many words as possible from the Slovak language.

For root segmentation, we utilized two lexical resources. The first dictionary from 2015, titled Retrograde Morphematic Dictionary of Slovak (Ološtiak, Genči & Rešovská, 2015), comprises over 70,000 morphemically segmented lexical units (lemmas), arranged in retrograde order. Root morphemes are bolded in the dictionary, grammatical morphemes are italicized, and all other types of morphemes are marked in standard font. The dictionary also includes additional annotations such as morphemic boundaries, information on derivational motivation, various markers capturing grammatical information, and many others, which are irrelevant for our research. An example of this dictionary is shown in Fig. 3. The English translations are indicated in blue.

Figure 3 An example of RDMS (Ološtiak, Genči & Rešovská, 2015).

The second dictionary we utilized was the Dictionary of Root Morphemes of Slovak (Sokolová, Ološtiak & Ivanová, 2012), which additionally includes some derived and alternative roots that were not present in RMSS.

After segmenting the roots for basic word forms (lemmas), we utilized the Morphological Database of the Slovak National Corpus (Slovak National Corpus, 2024), which contains over 98,000 unique lemmas with their corresponding full word paradigms (all inflected forms). In total, the database contains up to 1.19 million unique full word paradigms, along with tags capturing grammatical categories. Since we knew the roots for the lemma forms as we segmented them in the previous step, we were able to extract roots for other word forms from this morphological database. In total, we managed to extract roots for over a million words.

Text corpus

The aim of subword tokenization is to divide continuous text into smaller text units known as tokens. Current tokenization models utilizing BPE (Sennrich, Haddow & Birch, 2016) or WordPiece (Wu et al., 2016) algorithms are often trained on extensive text corpora. We decided to train the tokenizer using the OSCAR (Open Super-large Crawled Aggregated coRpus) text corpus (Ortiz Suárez, Sagot & Romary, 2019), which includes texts for up to 166 languages, including Slovak. We specifically used the 2019 version of OSCAR, which is derived from the Common Crawl corpus. We downloaded the Slovak part of this corpus using the HuggingFace library in Python.

The preprocessing of this corpus consisted of several steps. In the first step, we removed all text sections that were not in the Slovak language, as we aimed to train a purely Slovak tokenizer focusing on the morphology of the Slovak language. In the second step, various non-Slovak characters were removed, including mathematical symbols, pictograms, emoticons, and consecutive duplicate special and whitespace characters. In the third step, all duplicate and very similar sentences were removed, and URL addresses were replaced with the string “<url>”. After cleaning, the total size of the text corpus is 4.2 GB and it consists of a total of 589 million words, 3.7 million unique words, and contains only 222 unique characters.

Design and training of a morphological tokenizer

To ensure that word roots are not divided into multiple tokens and to preserve their integrity within individual tokens, it is essential for the roots to enter the BPE algorithm as indivisible units. There exists an implementation of the BPE training algorithm using the Tokenizers library from Hugging Face, which provides a powerful and flexible tool for text tokenization with support for various models and languages. However, it is not possible to enter the roots of words into this library's algorithm and indicate which roots should not be divided further. The implementation only allows input data in the form of text or file paths. Therefore, we decided to reprogram the BPE algorithm from scratch.

We started by loading the text corpus, which we normalized by converting all letters to lowercase. Then, we pre-tokenized the text using the GPT-2 tokenizer, which divides the input text into words with spaces and punctuation. Additionally, the GPT-2 tokenizer transforms the encoding into bytes, ensuring that the size of the basic vocabulary will be maximum 256 tokens and ensuring that every basic character will be included in the vocabulary.

Subsequently, we obtained the frequency of occurrence for all words and punctuation in the entire corpus (we created a dictionary called word_freqs). Finally, we added words to the list that were not found in the corpus but for which we had recorded roots.

When examining tokenization using the SlovakBERT tokenizer, we found that some words, especially the first words in a sentence without preceding whitespace, are often tokenized differently than the same words appearing within the sentence, i.e., with preceding whitespace. We decided to address this issue of ambiguous tokenization in our morphological tokenizer by adding whitespace before each valid word if it was not already present. We have already implemented this adjustment in the pre-existing word_freqs dictionary.

In the next step, we created the basic vocabulary, consisting of all characters used in the corpus (at the byte level), and added special tokens <s>, <pad>, </s>, <unk>, <mask>, which will be necessary during the training of the language model.

Next, it is necessary to prepare words for training the BPE tokenizer. In the original implementation, each word is divided into characters, but we divided each word into characters and roots if roots were recorded for that word. An example of word division (split dictionary) along with the frequency of occurrence can be seen in the following table (Table 1).

Table 1 Word segmentation into characters and roots with frequency of occurrence.

Word	Splitted word	Frequency	
“Ġrozhodol”	[“Ġ”, “roz”, “hod”, “o”, “l”]	49,581	
“ĠpredchÃ¡dzali”	[“Ġ”, “p”, “r”, “e”, “d”, “chÃ¡dz”, “a”, “l”, “i”]	1,421	
“Ġpsychotest”	[“Ġ”, “psych”, “o”, “test”]	150	

The training process itself involved iteratively iterating through individual words (in the word_freqs dictionary) and counting the frequencies of all pairs of adjacent tokens in the specific words from the splits dictionary. From these frequencies, we selected the pairs of tokens that were most commonly repeated and merged them into a single token. This merged token was added to the dictionary, and we replaced all occurrences of the corresponding pair of tokens in the splits dictionary. We repeated this process until we reached the maximum number of tokens, which we set to 50,264 (the same number as the SlovakBERT tokenizer).

The training of the tokenizer took approximately 180 h on an AMD EPYC 7542 32-Core Processor, which is several times longer compared to the optimized version of the implementation from Hugging Face, where it only took a few minutes. With the Tokenizers library, we also trained another tokenizer (pureBPE) on the same text as our morphological SKMT.

Results

The process of text tokenization using the trained tokenizer begins similarly to its training. The text is segmented into words, which are further divided into roots and individual characters. Subsequently, these tokens and merged pairs are used to create final tokens representing various parts of the text.

To evaluate the performance of the new tokenization method, we first compare the tokenization outputs of pureBPE, SKMT, and SlovakBERT. Then, we train models using pureBPE and SKMT tokenizers and evaluate their success in sentiment classification.

Tokenization performance evaluation

To evaluate the tokenization results of our method, we decided to compare the outcomes with tokenization using the SlovakBERT (SB) model tokenizer and with our custom pureBPE tokenizer, trained on the same text as our morphological tokenizer (SKMT). Even the SB tokenizer was partially trained on the same corpus.

For tokenization testing, we employed the OSCAR 2201 text corpus (Abadji et al., 2022). The entire Slovak portion of this corpus comprises 16.5 GB of data and contains 1.6 billion words. We randomly selected 500,000 text paragraphs from this corpus, ensuring that we chose paragraphs on which our tokenizers had not been trained. Each paragraph consists of approximately a few sentences. This sample seemed sufficiently representative and should ensure reliable tokenization results.

We evaluated several characteristics of tokenization, including the number of unique valid words (Word Count), the number of unique words for which roots were recorded (Word Root Count), the total number of tokens in the text (Total Tokens, TT), the average number of tokens per word (AT/W), the average number of tokens per rooted word (AT/RW), the average token length in characters for all valid unique words (Average Token Length, ATL), and the average token length in characters for unique words with recorded roots (Average Rooted Token Length, ARTL). Additionally, we assessed the percentage of correctly tokenized words with respect to roots (Correct Tokenization, CT).

For all acquired characteristics, we calculated basic statistics, which are provided in Table 2.

Table 2 Basic statistics of tokenization properties tested across three tokenizers.

	Mean	Std	Min	Q1	Q2	Q3	Max	
Word count	61.566	20.982	18.000	46.000	56.000	72.000	210.000	
Word root count	42.201	15.836	7.000	31.000	38.000	50.000	154.000	
SB TT	105.854	40.013	64.000	76.000	93.000	123.000	377.000	
SB AT/W	1.292	0.130	1.000	1.197	1.275	1.370	2.708	
SB AT/RW	1.276	0.129	1.000	1.184	1.261	1.353	2.235	
SB ATL	4.512	0.518	3.140	4.135	4.454	4.827	7.932	
SB ARTL	5.356	0.648	3.432	4.895	5.299	5.762	8.971	
SB CT	0.905	0.064	0.455	0.868	0.914	0.951	1.000	
pureBPE TT	100.097	38.206	46.000	72.000	88.000	117.000	350.000	
pureBPE AT/W	1.215	0.110	1.000	1.136	1.200	1.279	2.167	
pureBPE AT/RW	1.187	0.104	1.000	1.111	1.174	1.250	2.000	
pureBPE ATL	4.795	0.563	3.025	4.386	4.740	5.148	8.116	
pureBPE ARTL	5.749	0.682	3.591	5.260	5.694	6.189	9.400	
pureBPE CT	0.931	0.053	0.476	0.900	0.939	0.970	1.000	
SKMT TT	103.282	39.504	46.000	74.000	91.000	120.000	359.000	
SKMT AT/W	1.270	0.122	1.000	1.182	1.255	1.341	2.351	
SKMT AT/RW	1.229	0.101	1.000	1.157	1.220	1.290	2.061	
SKMT ATL	4.588	0.528	2.905	4.207	4.539	4.917	7.587	
SKMTARTL	5.551	0.632	3.460	5.098	5.492	5.950	9.132	
SKMT CT	0.997	0.010	0.807	1.000	1.000	1.000	1.000	

From the results of basic statistics, it is evident that the pureBPE tokenizer achieved the lowest values for characteristics such as TT, AT/W, and AT/RW, while also obtaining the highest values for the average token length in characters for both ATL and ARTL. Conversely, the SB tokenizer exhibited the highest values for TT, AT/W, and AT/RW, and the lowest values for ATL and ARTL, highlighting a clear contrast in performance between the two tokenizers. Additionally, while the SKMT tokenizer achieved the best results for the percentage of correctly tokenized words with respect to root integrity (CT), the SB tokenizer had the lowest performance in this characteristic.

For more detailed verification of the results, we decided to use box plots for all three tokenization methods: SB, pureBPE and SKMT.

Visual representations of the number of tokens can be observed in the form of box plot in Fig. 4. From the graph, it is clear that both tokenizers (pureBPE and SKMT) tokenize the text into a smaller number of tokens (TT characteristic) compared to SB, as the median for pureBPE (Q2 = 88) and SKMT (Q2 = 91) is smaller than for SB (Q2 = 93). Based on the minimum, lower quartile (Q1), median (Q2), upper quartile (Q3), and maximum, it can be confirmed that pureBPE tokenizes into the smallest number of tokens.

Figure 4 Box plot of the numbers of tokens.

A smaller number of tokens can potentially enhance the efficiency of a language model by reducing the computational operations required for text processing, thereby leading to faster training and inference. It can also improve the model’s understanding, as longer sequences may distract the model’s attention and hinder context capture. However, a smaller token count may occasionally be disadvantageous, resulting in loss of textual nuances and details that could affect model accuracy. The SKMT tokenizer strikes this balance, positioning itself in token count between pureBPE and SlovakBERT tokenizers.

Another characteristic we examined was the average number of tokens per word for all unique words (All Words-AT/W) and for unique words for which we know the roots (Only Rooted Words-AT/RW). The results from Fig. 5 suggest that both tokenizers (pureBPE and SKMT) achieve a lower average number of tokens per word compared to the SB tokenizer. Interestingly, the difference is even more pronounced for rooted words in both observed groups. Overall, it can be observed that pureBPE and SKMT consistently produce fewer tokens per word than SB, with SB showing the highest token count in this regard.

Figure 5 Boxplot of the average number of tokens per word.

Since the average number of tokens correlates with the average token length, it is not surprising that the results, depicted in Fig. 6, confirm the same trend as we found in the previous graph (Fig. 5). In this graph, the larger differences between the groups All Words (ATL) and Only Rooted Words (ARTL) can be seen even more clearly.

Figure 6 Boxplot of the average token length per word.

The most important results of our research can be observed from Fig. 7, which displays the percentage correctness of tokenized words concerning root integrity in tokens (CT characteristic). The correctness of a tokenized word concerning root integrity means that all roots within a single word will be segmented into tokens such that none of the roots are split across multiple tokens, but the entire root will be part of one token. This means that one token can contain not only the root but also other characters of the word that are before or after this root. It was already evident from Table 2 that the SB tokenizer yields the worst results (mean = 0.905, Q2 = 0.914), followed by pureBPE (mean = 0.931, Q2 = 0.939), with SKMT performing the best (mean = 0.997, Q2 = 1.000). These results are confirmed in this graph (Fig. 7).

Figure 7 Boxplot of the percentage of correctly tokenized words with respect to root morphemes.

For more thorough verification of the results, we decided to use residuals for this characteristic. We compare the tokenization results of pureBPE and SlovakBERT with the tokenization of SKMT. This comparison differs from the previous one in that we now take the percentage difference between SB and SKMT (denoted as R1) and pureBPE and SKMT (denoted as R2) for specific tokenized texts, not just calculated statistics for individual tokenizers.

The results were confirmed by this approach as well (Fig. 8), as Q1 (−0.128), Q2 (−0.083), and Q3 (−0.047) for the residuals R1 (SB-SKMT) are less than 0, indicating that the SB tokenizer tokenizes words with lower percentage correctness concerning root integrity compared to SKMT. A similar pattern is observed for the residuals R2 (pureBPE-SKMT). Since Q1 (−0.095), Q2 (−0.059), and Q3 (−0.029) are also less than 0, it is clear that the SKMT tokenizer tokenizes words with higher percentage correctness concerning root integrity compared to the pureBPE tokenizer. Based on the measured values of R1 and R2, it can be concluded that the SKMT tokenizer is more accurate in preserving root integrity than pureBPE and SB.

Figure 8 Boxplot of the residuals of percentage of correctly tokenized words with respect to root morphemes.

Tokenization evaluation using language modeling

Since our new tokenization method, SKMT, achieved the best results based on the average root integrity value in the resulting tokens, we decided to validate this tokenizer on a language model. We used the RoBERTa architecture (Liu et al., 2019), which is based on the Transformer architecture (Vaswani et al., 2017), for pretraining the models. One model was trained using the pureBPE tokenizer (named SK_BPE_BLM – SlovaK BPE Baby Language Model), and the other was trained using our SKMT (named SK_Morph_BLM – SlovaK Morphological Baby Language Model). Our model consisted of six layers of encoders, 12 attention heads, a hidden size of 576, and a total of 58 million parameters. The maximum position embeddings were set to 258, corresponding to the maximum sequence length. The model was trained with a batch size of 128 for approximately 10 epochs. We used the Adam optimizer with a learning rate of 1×10−4, a dropout rate of 0.1, and a weight decay of 0.01. We pretrained these models on the same text used for evaluating the tokenization. It is important to emphasize that our goal was not to create a model that surpasses current models containing billions of parameters, but to demonstrate that our proposed tokenization method (SKMT) improves language modeling compared to the currently used BPE method.

We evaluate the language modeling itself using the perplexity metric, which measures how well the model predicts a masked token in a sequence. It is calculated as the exponential value of the cross-entropy (L-average logarithmic loss) of the model. The lower this value, the better the model is at predicting the correct tokens, indicating a better understanding of the text (Le Bronnec et al., 2024).

(1) Perplexity=eL

The data (Fig. 9) indicate that the model trained with the SKMT tokenized text (SK_Morph_BLM) outperformed the model trained with the BPE tokenized text. Specifically, the SKMT model achieved a lower evaluation loss with a median of 0.187 and a corresponding lower perplexity with a median of 1.206 at the conclusion of the pretraining phase (epoch 9). In contrast, the BPE model demonstrated an evaluation loss with a median of 0.251 and a perplexity with a median of 1.285. These results strongly suggest that the SKMT method facilitates a more nuanced understanding of linguistic structures compared to the traditional BPE approach. Further validation of these findings will be conducted through an NLP task focused on sentiment classification.

Figure 9 Evaluation loss over epochs.

For fine-tuning the language models on the NLP task of sentiment classification, we utilized the Reviews dataset, which was created by the authors of the SlovakBERT model (Pikuliak et al., 2022) for the purpose of fine-tuning and evaluating this model. This dataset comprises reviews from various domains: accommodation, books, cars, games, mobiles and movies. In total, the dataset consists of 677 entries labeled with sentiments: negative (315 entries), neutral (57 entries), and positive (305 entries). Preprocessing involved manual correction of diacritics, which were missing in some entries. The text from the dataset was tokenized using both tokenizers, and the models were evaluated through fine-tuning using stratified 10-fold cross-validation. We reduced the learning rate to 1×10−5 and set the batch size to 16. A classification layer was added to the model, which determines the classification into one of the three sentiment categories.

Fine-tuning was conducted over 10 epochs across all 10 folds. After each epoch, basic evaluation metrics were calculated: accuracy, precision, recall, and F1-score. The result of each fold was the highest F1-score achieved. In Fig. 10, we present the results of this validation, showing the metrics for accuracy and F1-score. The primary metric for evaluation will be the F1-score, due to the imbalance in the dataset. From the graph, it is evident based on the F1-score metric that the SK_Morph_BLM model (median = 0.787) was able to classify sentiment better than the SK_BPE_BLM model (median = 0.753) by nearly 3.5%.

Figure 10 Comparison of accuracy and F1 score metrics across 10 folds for models.

To examine the results in greater detail, we generated a confusion matrix (Fig. 11) that displays the correct and incorrect predictions for each category, combined in a single graph across all folds. For negative sentiment, the SK_Morph_BLM model performs 1% worse compared to the SK_BPE_BLM model. However, for neutral and positive sentiments, SK_Morph_BLM significantly outperforms SK_BPE_BLM, specifically by 14.1% for neutral sentiment and by 3.9% for positive sentiment.

Figure 11 Confusion matrices comparing sentiment classification performance of models across all folds.

Following the methodology outlined in the thesis “Distilling the Knowledge of SlovakBERT” (Agarský, 2022), we focused on the Semantic Textual Similarity (STS) task after fine-tuning our language models on sentiment classification. For this task, we utilized the Semantic Textual Similarity Benchmark (STS-B) dataset (Cer et al., 2017), which we translated from English into Slovak. This benchmark consists of sentence pairs from various domains and genres, rated for their similarity on a scale from 0.0 to 5.0. In the original version of the dataset, these sentence pairs were divided into training, development, and test sets. We combined the training and development sets, excluding the test set that did not contain similarity ratings, resulting in a total of 7,163 sentence pairs with similarity ratings. The similarity ratings were normalized to a scale of 0.0 to 1.0.

To fine-tune and evaluate the models for this task, we again employed stratified 10-fold cross-validation. Each fold was trained for a maximum of 15 epochs, implementing an early stopping mechanism based on the mean squared error (MSE) metric to prevent overfitting when performance no longer improved. We set the learning rate to 1×10−5 and the batch size to 32. A regression layer was added to the model to predict the continuous value of semantic similarity.

For evaluating text similarity, we used the Pearson and Spearman correlation coefficients, which provide a comprehensive view of the model’s quality in measuring the similarity between sentence pairs. In Fig. 12, we present the STS evaluation results, showing the Pearson and Spearman coefficients. The graph clearly indicates that the SK_Morph_BLM model achieved higher median values of Pearson (Q2 = 0.612) and Spearman (Q2 = 0.586) coefficients compared to the SK_BPE_BLM model, which achieved median values of Pearson (Q2 = 0.512) and Spearman (Q2 = 0.501) coefficients. These results suggest that the SK_Morph_BLM model, which is based on morphological tokenization, better captures the similarity between sentence pairs than the SK_BPE_BLM model.

Figure 12 Box plot of STS evaluation results: Pearson and Spearman correlation coefficients.

Discussion

The only limitation of this research we see is that the SlovakBERT tokenizer was trained on text that was not case-normalized. Therefore, this tokenizer tokenizes some words differently if the word contains uppercase letters. We noticed that this tokenizer, along with others, tokenizes the first words in a sentence differently than if the same word appeared within the sentence. This is because the first word does not have a space in front of it. We addressed this ambiguity in tokenization with our SKMT by automatically inserting a space before each valid word.

By evaluating the results, we have demonstrated that it is possible to create a more efficient tokenizer using root morphemes, with a minor improvement observed in STS task. We believe that the proposed method could be applied to most inflected languages. The most challenging part was undoubtedly the extraction of root morphemes. We decided not to use any library that could automatically extract roots from words. Instead, we used morphological dictionaries compiled by linguists, ensuring the complete reliability of root determination.

Conclusions

In this contribution, we introduce a novel approach to text tokenization based on the morphology of the Slovak language and the Byte Pair Encoding algorithm. Our method, named SlovaK Morphological Tokenizer (SKMT), achieved significant results in preserving the integrity of word roots within individual tokens. We leveraged the original BPE algorithm, where the word roots themselves entered the algorithm as indivisible units. The word roots were extractable with the aid of a morphological database and two morphemic dictionaries created through manual linguistic efforts.

Comparison with the existing SlovakBERT tokenizer, as well as our trained BPE tokenizer (pureBPE), demonstrated that our SKMT achieves more accurate results in preserving the integrity of word roots. Specifically, we found that SKMT achieved an average accuracy of word tokenization with respect to root integrity at 99.7%, whereas SlovakBERT achieved only 90.5% and pureBPE 93.1%.

Further validation involved fine-tuning models on a sentiment classification NLP task, where models trained with SKMT achieved an F1-score improvement of 3.5% over those trained with conventional BPE tokenization. The STS evaluation results demonstrate that the SK_Morph_BLM model, based on morphological tokenization, outperformed the SK_BPE_BLM model. The SK_Morph_BLM model achieved higher median values for both the Pearson (Q2 = 0.612) and Spearman (Q2 = 0.586) coefficients, compared to the SK_BPE_BLM model, which recorded median values of Pearson (Q2 = 0.512) and Spearman (Q2 = 0.501). These findings indicate that the SK_Morph_BLM model more effectively captures the similarity between sentence pairs.

These results suggest that our approach could have significant applications in the development of language models and natural language text processing, leading to better understanding and precision. In the future, we aim to expand our research and attempt to create a morphological tokenizer without byte encoding, which could potentially further enhance tokenization.

In the future, we aim to expand our research by evaluating these models across multiple NLP tasks to comprehensively explore the differences between tokenizers.

Supplemental Information

Supplemental Information 1 Supplemental Materials.

Additional Information and Declarations

Competing Interests

Author Contributions

Data Availability

The authors declare that they have no competing interests.

Dávid Držík conceived and designed the experiments, performed the experiments, analyzed the data, performed the computation work, prepared figures and/or tables, and approved the final draft.

Frantisek Forgac performed the experiments, authored or reviewed drafts of the article, and approved the final draft.

The following information was supplied regarding data availability:

The tokenizers are available at GitHub and Zenodo:

- https://github.com/daviddrzik/Slovak_subword_tokenizers.

- Dávid Držík. (2024). daviddrzik/Slovak_subword_tokenizers: Slovak subword tokenizers (v1.0). Zenodo. https://doi.org/10.5281/zenodo.13910796

The tokenizers and pre-training and fine-tuning models are available in the Supplemental Files.

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
