# Peer review of "Slovak morphological tokenizer using the Byte-Pair Encoding algorithm"

_PeerJ Computer Science, doi:10.7717/peerj-cs.2465_

## Round 0.1 · original submission · Major Revisions

Dear Dávid Držík and Frantisek Forgac,

Thank you for submitting your manuscript, "Slovak morphological tokenizer using the Byte-Pair Encoding algorithm," to PeerJ. After careful consideration and a thorough evaluation by our reviewers, we have decided that your manuscript can be reconsidered for publication after major revisions.
The reviewers have raised significant concerns regarding the methodological aspects of your study, particularly the evaluation metrics and experimental design. Below is a summary of the essential revisions needed:
1. Comparison with Existing Methods: Enhance the literature review to include a comparison with other tokenization methods like WordPiece and SentencePiece. This should not only discuss the methods but also justify the choice of BPE for your study.
2. Clarification of Evaluation Metrics: The metrics used to evaluate the tokenizer need to be clearly defined and justified. It is crucial to explain how these metrics reflect the performance and utility of the tokenizer in language modeling.
3. Experimental Design and Data Presentation: Revise the experimental design to provide a clearer, more logical flow of information. Improve the presentation of your results, particularly in Table 3. Consider using side-by-side box plots or other visual aids to make the data more accessible.
4. Methodological Rigor: Address the methodological issues pointed out, such as the need for a more robust statistical analysis and clarity in the methods used to assess tokenization accuracy.
5. Broader Implications and Comparisons: Provide a more detailed discussion on how your tokenizer compares with standard models like SlovakBERT in practical applications, not just in maintaining root integrity.
6. Reproducibility and Open Science: Ensure that all materials required for replicating your experiments, such as input files and detailed methodology, are clearly listed and accessible.

To move forward, it is essential that you address these concerns comprehensively in your revision. Please refer to the detailed comments from each reviewer, as enclosed, and respond to each point explicitly in your revised manuscript.

Please note that this decision does not guarantee acceptance of your manuscript; the revised version will undergo another round of review. We appreciate the effort you have put into your manuscript and believe that with careful revision, it can potentially meet the publication standards of PeerJ.
We look forward to receiving your revised manuscript.

Reviewer 1 ·

Basic reporting

The paper proposes a tokenization method for the Slovak language that preserves word roots.
The overview part focuses on BPE tokenization. It does not contain references to other tokenization methods that are currently used with large language models - WordPiece and SentencePiece, or previous unsupervised tokenization approaches, such as Morfessor.

Experimental design

The experiments compare the proposed modification of the BPE algorithm with another implementation - pureBPE and an already trained BPE tokenizer from the SlovakBERT language model.

The evaluation methodology is not correct.
As an evaluation metric, authors count the “percentage of successfully tokenized words concerning root integrity” (L281), In the abstract it is called “accuracy”.
“To evaluate the tokenization results of our method, we decided to compare the outcomes with
tokenization using the SlovakBERT” (275 ). It does not make sense to measure the difference with the SlovakBERT tokenizer.

They also do not describe how they count correct and incorrect tokenization. Does it mean that good tokens are the same as word roots? Or does it mean that “successful” tokens are larger than word roots because the word root is not split?

The other statistics do not say anything about the “quality of tokenization” as well.
L282: “counts of unique words, total number of tokens, average number of tokens per word, and average token length in characters for all unique words, as well as for unique words for which we recorded roots”

Validity of the findings

Authors try to show that the proposed tokenization method preserves “word roots”, but it does not imply that it improves language modeling. The paper or the referenced literature does not support this claim.
If the authors claim that the proposed tokenization improves language modeling, they should show how.

The proposed tokenization approach is based on the Dictionary of Root Morphemes of Slovak (Sokolová, Ološtiak & Ivanová, 2012) Retrograde Morphematic Dictionary of Slovak (Ološtiak, Genči, Rešovská, 2015), which are not publicly available. Therefore, it might be complicated to repeat the experiments.
The identified word roots are considered unbreakable symbols for the BPE tokenization algorithm.
Texts for further training of the modified BPE tokenizer come from the Slovak part of the OSCAR corpus.

Additional comments

The main objection against this paper is that it is missing experiments with language models.

The second important issue is with the evaluation methodology.

The proposed tokenization algorithm has the following problems that should be addressed in the paper:
Preserving word roots from tokenization might lead to lower robustness against spelling errors.
The tokenizer focuses only on the word roots. It does not cover the full properties of the Slovak morphology, such as prefixes and irregular suffixes.

The bright side of the paper is that the topic is interesting and the proposed ideas are worth further research. If the tokenizer improves the performance of the language model, then it is possible without increasing computational complexity. Proper tokenization of the text enhances the context size that the language model can handle in one batch.

Cite this review as

·

Basic reporting

This paper describes a relatively straightforward method for
augmenting byte-pair-encoding (BPE, a popular tokenization method for
pretrained language models) to preserve the root forms of the words.
The method is compared against an non-augmented version of the
algorithm. Overall, the inquiry seem worthwhile. My criticism centers
mainly around the evaluation of the approach.

- The paper is written in a clear English,

- The figures/tables are formatted reasonably, but the presentation of
results is very difficult to follow. The main table (Table 3) is
already very crowded where differences are not easy to extract for
human eyes. Having the meaning of symbols defined in another table
makes it even more difficult to follow. I suggest at least making
the names of the evaluation metrics more meaningful and move the
information in Table 2 to the caption of Table 3. However, I
strongly suggest presenting these as side-by-side box plots (they
are exactly made for making this type of distribution information
easy on human eyes).

- Some of the terms/measures used may benefit from clearer
definitions.

- The paper would benefit from reviewing some of the work on
unsupervised morphology learning / segmentation (more on this
below).

Experimental design

The topic of the paper, as far as I can tell, is appropriate for the
journal, the research question is well defined, and motivated
reasonably well. Experiments seem to be performed well, the source
code and resulting data are provided (I did not check them carefully),
however, I do not see the input files (word lists extracted from the
lexicons) which is crucial for replication.

Validity of the findings

This is where most of my criticism goes. The concrete points are
listed below with approximately ordered by importance:

- I think the evaluation metrics ("characteristics") are not fit for
evaluating the approach. The point from the start is keeping the
root integrity which should preserve the semantics of the roots, and
relieve the models from composing seemingly meaningless tokens
(Keeping root integrity hopefully may also mean the non-root subword
tokens are also meaning preserving). So, it is obvious that the root
injection into the tokenization will improve the "correctly
tokenized words with respect to roots". And, all other metrics favor
the typical BPE objectives, so it is no surprise that pureBPE is
good at them. In short: none of the "characteristics" tells us the
objective is achieved.

Short of doing computationally expensive pretraining experiments one
needs evaluation metrics that do not state the obvious, but tap into
the meaning preserved with the proposed tokenization algorithm. I do
not have a concrete suggestion, but perhaps trying these on
small-scale models may be an option (e.g., sort of models like
BabyBERTa <https://aclanthology.org/2021.conll-1.49/>). Then, it is
rather obvious that the pureBPE will be more optimum with respect to
its own objective (shorter sequences with fixed vocabulary size).
So, one should document the cost of this non-optimal segmentation
(e.g., on average the increase of sequence length, which is likely
to increase the computation needed).

- Since the method seems to aim for more "linguistically sound"
tokenization, I'd expect some review of the earlier work on
unsupervised morphology learning. And, perhaps comparing with an
off-the-shelf tool like Morfessor that already shares similar aims.

- The comparison with the SlovakBERT tokenizer is not sound. As noted
in the manuscript, there is a difference of case normalization
between the tokenizers trained in this study, and the SlovakBERT
tokenizer.

- Related to above, I do not find the "residual" metric (the
difference between the tokenizers and the SlovakBERT tokenizer)
proposed in any way more intuitive. In fact, I'd appreciate absolute
numbers for each of the metrics, most of which are quite
interpretable. If there is anything I am missing about the
"usefulness" of the residuals, it should be explained.

- Around line 184, the reader encounters "five characteristics" that
are used for evaluation, but since these characteristics are
important for the argument, I think they should be introduced and
motivated (why are they good measures of success for a tokenizer?)
early in the paper.

Additional comments

Some minor (mostly language/style/clarification) suggestions:

- lines 31-32: the main motivation of (subword) tokenization is not to
"transform[ing text] into a numerical format", but to provide the
model with a sequence containing "meaningful" units. "Numerical
format" is a given for representing any object (not necessarily) for
machine-learning methods. Further, the motivation of subword
tokenization is to limit the size of the vocabulary without missing
out too much on the meaning side. So, it may be worth noting here.

- BLOOM citation is inaccurate, the first author shows up as "Workshop".

- Line 206: something went wrong with LaTeX formatting.

- Line 224: I suggest including text size in more meaningful units
like tokens, lines, documents rather than GB.

- Also on line 279: the unit "sections" needs clarification.

- Line 264: The computing time is not meaningful without specifying
the hardware the implementations were run on.

- Line 294: although I strongly recommend abandoning it in favor of
absolute values, if the "residual" evaluation is kept, it should be
clear at first mention what it means. It is not the obvious use of
the term.

- Figure 1 is not needed. This can be made a simple in-text example
(perhaps a numbered example as it is the convention in the field of
linguistics).

- Most figures leave the description of the abbreviations out. These
should be placed in the caption. I do not know if this is journal
policy, but I would also appreciate a lot if the figures/tables were
inserted closer to their first reference.

- All non-English examples should include glosses/translations - even
if this may look not necessary for understanding their point.

- Many bibliography items have missing information (mostly place of
publication, but a thorough check is recommended).

Cite this review as

Reviewer 3 ·

Basic reporting

The text combines elements typical of a diploma thesis with phrases that might be found in a conversation with ChatGPT. The literature review is structured as a timeline focusing on the development of the Byte-Pair Encoding (BPE) algorithm, though it lacks depth in its analysis.


For instance, line #112 states: "Based on their results, Bostrom & Durrett (2020) advocate for a shift in the standard practices of language model pretraining. They recommend that NLP researchers and practitioners consider adopting unigram LM tokenization over BPE, citing its superior performance and better alignment with linguistic structures. This recommendation is poised to influence future research directions and pretraining practices in the NLP community."

Additionally, there are instances of incomplete or nonsensical sentences, such as those found in lines #353-355.
The formatting of mathematical expressions also requires attention, suggesting the use of tools such as Microsoft Word’s equation editor or LaTeX commands to ensure clarity and professionalism in the presentation of complex notations (e.g. \mathcal{O}(NM), \mathcal{O}(N \log M), etc.).

In reviewing the figures included in the manuscript, there are concerns about why figures are not embedded directly in the pdf document and are instead accessed as separate supplementary files. This format disrupts the reader's engagement and complicates the process of referencing visual data. Moreover, the figures lack detailed captions and corresponding text explanations. For instance, Figure 1 does not clarify which tokenizer's output is depicted, which is crucial for understanding the context and relevance of the visual data presented. Incorporating figures directly into the document with comprehensive captions and clear text references would significantly enhance the clarity and effectiveness of the paper.

Experimental design

The authors claim that their article makes significant contributions to optimizing preprocessing steps for model training. However, the paper should explicitly enumerate these contributions in a clear and structured manner, such as using bullet points. Currently, the document contains vague promises, for instance, mentioning the generation of language vectors similar to typological language vectors like lang2vec, which are used in various NLP applications. This claim, among others, lacks substantiation.

Further, while the authors suggest that their approach could influence the field similarly to how BERT-like embeddings demonstrate typological clusters in their embedding spaces, as detailed in existing literature, these assertions are not yet backed by conclusive evidence or comparative analyses within the paper. For a more robust presentation, the authors should provide detailed evidence supporting their claims, align their findings with established research, and clearly articulate their novel contributions in relation to existing methods and results. This would not only clarify the actual advancements made but also place their work within the broader context of ongoing research in NLP.

The paper's primary contribution appears to be a stemming algorithm, but the authors do not present evidence of its efficacy in practical NLP applications. Specifically, they fail to demonstrate through downstream tasks how their tokenization strategy enhances performance in any specific NLP application. For a more compelling and complete academic contribution, it would be beneficial for the paper to include experimental results or case studies that show the tangible benefits of this stemming approach over existing methods, particularly in real-world NLP tasks such as sentiment analysis, machine translation, or text classification. Without this critical evaluation, the practical relevance and impact of the proposed method remain unclear.

Validity of the findings

The core issue here is whether maintaining root integrity is the most significant contribution of SKMT, or if it is being overemphasized in the absence of broader performance metrics across diverse NLP tasks.

Comparative Basis and Relevance: The comparison of SKMT’s ability to maintain root integrity with models not specifically trained for this metric (like SlovakBERT and a native BPE tokenizer) could potentially skew the perceived effectiveness of SKMT. Since these models have different primary objectives, SKMT naturally might outperform them in maintaining root integrity but this does not necessarily translate to overall better performance in practical NLP applications.

Root Integrity as a Metric: While root integrity might be a valuable metric for specific linguistic or lexicographic applications where the preservation of morphological properties is crucial, it’s not universally applicable as a measure of tokenizer effectiveness across all NLP tasks. Other aspects such as context preservation, token distribution, and adaptability to varied linguistic inputs are also important.

Experimental Design and Validation: To solidify the claims made about SKMT, the paper should detail the methodology used for calculating root integrity. Was it a simple binary method assessing the presence or absence of roots, or a more complex calculation like edit distance or another form of similarity measurement? Moreover, demonstrating how maintaining root integrity impacts the performance of downstream tasks (e.g., through ablation studies or comparative analyses with tasks where root integrity could play a crucial role) would provide a more rounded evaluation of the tokenizer's utility.

The manuscript mentions selecting only 20% of text sections from the original corpus, totaling 895,125 sections, for some aspect of the study (line #278-279). It’s crucial to clarify if these sections were exclusively used for testing and excluded from training data, as mixing these can affect the validity of the results.

Regarding the comparison of tokenizers, the paper states that pureBPE outperformed others in all characteristics except for root integrity (line #289-291). However, the document fails to specify what these 'all characteristics' are. Moreover, terms like Q2, SB0-SBB5, and pureBPE1-5 are not defined, making it difficult to understand the basis of comparison and results.

The assertion that the SB tokenizer yields the worst results is vague and lacks context (line #315). It’s essential to specify the tasks or metrics where the SB tokenizer underperformed. Since no downstream NLP tasks were included in the study to assess the overall effectiveness of these tokenizers within a language model, making a broad comparison or ranking of the tokenizers seems premature.

Finally, the paper claims that the only limitation of the research is the non-normalization of case in the SlovakBERT tokenizer's training data (line #327). This statement overlooks other significant limitations, such as the treatment of cased tokenizers and the lack of evaluation in standard benchmark tests like text classification or named entity recognition. These omissions suggest that the new tokenizer’s utility within an NLP pipeline remains unproven, which is a substantial limitation that should be acknowledged.

Additional comments

In line #206, the manuscript mentions "98,000 lem full word paradigms." It is unclear whether this refers to 98,000 unique lemmas. Clarification is needed to understand precisely what is meant by "full word paradigms."

Concerning the dataset, the paper states that "the total size of the text corpus is 4.2 GB" (line #224). However, additional information about the dataset is necessary for a thorough evaluation. It would be beneficial to provide basic descriptive statistics of the dataset, such as the number of documents, average document length, and token count. Furthermore, details about the source of the data—whether it comes from domains like Wikipedia, tales, or news articles—would significantly enhance understanding. Additionally, it would be helpful to know if there are plans to publish the dataset for broader use in the research community, as well as for reproducibility purposes.

Regarding tokenization, the observation in lines #244-245 about the first words in sentences being tokenized differently when they appear without preceding whitespace raises an interesting point. Rather than viewing this as a mere issue, it's worth considering if this behavior might reflect contextual variations in language use, which could be an intentional feature of the tokenizer. A more in-depth analysis of this behavior could provide valuable insights into the tokenizer's design and functionality.

Cite this review as

---

## Round 0.2 · Major Revisions

Dear authors:

Thank you for your submission. After careful consideration, it has been decided that your manuscript titled "SlovaK Morphological Tokenizer (SKMT): Integrating Morphology into Tokenization" requires major revisions before it can be considered for publication.

Reviewer 3 has provided detailed feedback that we believe will be crucial in enhancing the quality and clarity of your work. Key points include:

1. Improve clarity and accuracy in reporting, especially in the introduction and citation linking.

2. Address methodological concerns for clarity and reproducibility, including data separation and open-sourcing the code.

3. Provide a deeper exploration of the findings, particularly regarding the Total Tokens characteristic and its impact on model performance.

4. Discuss the practical applicability and broader validation of the SKMT tokenizer.

Please address these points thoroughly in your revised manuscript. We look forward to your resubmission.

Reviewer 3 ·

Basic reporting

The paper titled "SlovaK Morphological Tokenizer (SKMT): Integrating Morphology into Tokenization" is generally well-structured, but there are areas where the reporting can be improved for clarity and accuracy. The use of future tense in the introduction is inappropriate for a scientific paper, which usually demands a more assertive tone about the research already conducted. For example, instead of "we will do that," a more appropriate phrasing would be, "This study aims to achieve that." Furthermore, the citation of Gutierrez-Vasques, Bentz, and others is not linked to the references section, making it difficult for readers to locate the source material. Similarly, the article lacks citations for key resources, including the RoBERTa model and the reviews dataset used (link: https://arxiv.org/abs/2109.15254), making it crucial to include these to maintain scholarly rigor. Additionally, the original STSB-sk report results, published by Ivan Agarsky and referenced by Richard Cepka, showed outcomes comparable to the SK_Morph_BLM model, which should be highlighted as a baseline to contextualize improvements in performance.

Experimental design

The experimental design of the paper presents a novel approach to text tokenization by introducing SKMT, which integrates Slovak language morphology into the Byte-Pair Encoding (BPE) algorithm. However, some methodological concerns need to be addressed for clarity and reproducibility. The authors mentioned selecting 20% of text paragraphs for tokenization testing from the original corpus, which may raise concerns about data leakage if these paragraphs were part of the training data. It is a fundamental principle of experimental design that test data should be entirely separate from training data to ensure valid and unbiased evaluation of the model's performance. Additionally, there is no clear indication of whether the authors plan to open-source the tokenizer or the trained models. Open-sourcing the code and models would enhance reproducibility and allow other researchers to validate the findings, contributing to a more robust scientific community.

Validity of the findings

The findings presented in the paper indicate that the SKMT tokenizer significantly enhances the integrity of word roots and improves the performance of language models on a sentiment classification NLP task. The authors report a remarkable 99.7% root integrity with SKMT compared to SlovakBERT and a native BPE tokenizer. Additionally, there is an improvement in F1 score by 3.5% over conventional BPE tokenization methods. However, it is unclear how the authors interpret the Total Tokens (TT) characteristic and whether a higher or lower number of tokens is desirable for optimal performance. A deeper exploration of this characteristic is essential to understand why fewer tokens per word might be advantageous or detrimental to a model's performance. The paper suggests an advocacy for a minimal number of tokens per word, but without a detailed discussion or empirical evidence on why this is beneficial, it is challenging to fully validate the impact of this approach. Additionally, while the authors claim improvements over their baseline, it is important to note that these results have not yet been reflected on widely recognized leaderboards for the specific NLP tasks discussed. The absence of leaderboard results raises questions about the practical significance and broader applicability of the SKMT tokenizer, suggesting that further validation in diverse contexts is necessary to substantiate the claimed performance gains.

Additional comments

While the SKMT tokenizer presents an interesting development in the field of tokenization, its practical applicability should be discussed further. Tokenization is a critical step in NLP that directly impacts the performance of language models; however, a tokenizer alone cannot determine a model's success. Instead, its utility is measured by the extent to which it can enhance a language model's performance on specific tasks. Additionally, it would be beneficial for the authors to explore which characteristics (Total Tokens, Average Tokens per Word, etc.) are particularly relevant to different segments of the NLP community. Providing concrete examples or potential applications of SKMT's improved tokenization could offer insights into how these characteristics might be leveraged to benefit various NLP tasks. Lastly, enhancing transparency by releasing the tokenizer and trained models publicly would facilitate further exploration and validation within the research community, ensuring the reproducibility of the study's claims.

Cite this review as

---

## Round 0.3 · accepted · Accept

The authors have addressed all the suggestions by the reviewers.

Congratulations!

· Appeal

Appeal


· · Academic Editor

Reject

Dear authors,

Thank you for submitting your manuscript and for your continued efforts in revising it. After careful consideration and review of the most recent submission, we regret to inform you that we are unable to proceed with the publication of your work.

Despite the revisions, the manuscript still does not meet the necessary standards and expectations of our reviewers.

Specifically, some of the key experiments requested have not been adequately addressed, which has impacted the overall quality and completeness of the research.We appreciate your understanding of this decision and thank you again for considering our journal for your work.

We wish you the best of success in your future research endeavors.

Sincerely,
José Alberto Benítez Andrades

Reviewer 4 ·

Basic reporting

The article reports the Slovak Morphological Tokenizer (SKMT), which enhances tokenization by incorporating Slovak language morphology into the Byte-Pair Encoding (BPE) algorithm. It preserves the integrity of word roots. The performed evaluations have shown SKMT achieves superior root integrity (99.7%) compared to SlovakBERT (90.5%) and pureBPE (93.1%). Models fine-tuned with SKMT on sentiment classification tasks yielded a 3.5% F1-score improvement.

- Professional English used along the manuscript seems clear and unambiguous.
- The referenced literature seems appropiate and sufficient.
- The article structure, figures and tables is acceptable. Raw data and code is also shared.
- The submission represents a unit of publication with self-contained relevant results.

Experimental design

The experimental design seems fair to me. Methods are detailed with sufficient detail to be replicated. The investigation is rigorous and complying with ethical and technical standards.

Validity of the findings

The findings seem valid to me. It is very impressive that SKMT's has a 99.7% root integrity. **

The discussion section might be a little bit two short when compared to the results section though. Authors could elaborate and relate a bit more their outcomes with previous works.

Cite this review as